# Crystal Structures of the *Clostridium botulinum* Neurotoxin A6 Cell Binding Domain Alone and in Complex with GD1a Reveal Significant Conformational Flexibility

**DOI:** 10.3390/ijms23179620

**Published:** 2022-08-25

**Authors:** Kyle S. Gregory, Anna R. Newell, Otsile O. Mojanaga, Sai Man Liu, K. Ravi Acharya

**Affiliations:** 1Department of Life Sciences, University of Bath, Claverton Down, Bath BA2 7AY, UK; 2Protein Sciences Department, Ipsen Bioinnovation Limited, 102 Park Drive, Milton Park, Abingdon OX14 4RY, UK

**Keywords:** botulinum neurotoxin, cell-binding domain, subtype A6, crystal structure, ganglioside binding, redox switch, conformational flexibility

## Abstract

*Clostridium botulinum* neurotoxin A (BoNT/A) targets the soluble *N*-ethylmaleimide-sensitive factor attachment protein receptor (SNARE) complex, by cleaving synaptosomal-associated protein of 25 kDa size (*SNAP**-**25*). Cleavage of SNAP-25 results in flaccid paralysis due to repression of synaptic transmission at the neuromuscular junction. This activity has been exploited to treat a range of diseases associated with hypersecretion of neurotransmitters, with formulations of BoNT/A commercially available as therapeutics. Generally, BoNT activity is facilitated by three essential domains within the molecule, the cell binding domain (H_C_), the translocation domain (H_N_), and the catalytic domain (LC). The H_C,_ which consists of an N-terminal (H_CN_) and a C-terminal (H_CC_) subdomain, is responsible for BoNT’s high target specificity where it forms a dual-receptor complex with synaptic vesicle protein 2 (SV2) and a ganglioside receptor on the surface of motor neurons. In this study, we have determined the crystal structure of botulinum neurotoxin A6 cell binding domain (H_C_/A6) in complex with GD1a and describe the interactions involved in ganglioside binding. We also present a new crystal form of wild type H_C_/A6 (crystal form II) where a large ‘hinge motion’ between the H_CN_ and H_CC_ subdomains is observed. These structures, along with a comparison to the previously determined wild type crystal structure of H_C_/A6 (crystal form I), reveals the degree of conformational flexibility exhibited by H_C_/A6.

## 1. Introduction

*Clostridium botulinum* neurotoxin (BoNT) is renowned as the most potent toxin known to humans [1]. It is the causative agent of botulism; thankfully, outbreaks of this deadly disease are incredibly rare [2]. Botulism causes flaccid paralysis by inhibiting acetylcholine release at the neuromuscular junction (NMJ) due to cleavage of a SNARE (soluble *N*-ethylmaleimide-sensitive factor attachment protein receptor) protein required for neurotransmitter release. Considering the low incidence of botulism, there has been no need for mass vaccination against BoNT; indeed, such measures would be undesirable [3] due to its increasing use as a therapeutic for the treatment of hyper-muscular and glandular disorders [3,4,5]. Historically BoNT has been categorised into seven immunologically distinct serotypes (BoNT/A-/G), however, with the emergence of mosaic (BoNT/DC, /CD, /FA, /HA) and BoNT-like proteins (BoNT/Wo and BoNT/En [6,7,8]) guidelines on BoNT nomenclature have been introduced to limit confusion within the literature [9]. The serotypes are further divided into subtypes due to minor differences in amino acid sequence, which have been associated with significant variation in toxicity across BoNT/A subtypes [10,11,12,13,14].

BoNT is expressed by *Clostridium botulinum* as a single polypeptide from a *bont* gene cluster [15] and is cleaved post-translationally by either a host or an endogenous protease into the active di-chain [16,17]. Non-toxic-non-hemagglutinin protein (NTNH) and neurotoxin associated proteins (NAPs) are co-expressed with BoNT and together they form the progenitor neurotoxin complex, which protects BoNT during its passage through the digestive system and into the bloodstream [18]. BoNT consists of three domains each with a specific role in the mechanism of toxicity. Firstly, the cell binding domain (H_C_), which consists of an N-terminal (H_CN_) and C-terminal (H_CC_) subdomain, utilises the H_CC_ to bind to both a protein (e.g., SV2 for BoNT/A) and a ganglioside (e.g., GT1b or GD1a) on the surface of motor neurons. BoNT is then internalized into an endosome via the endocytic pathway where the acidic environment is believed to cause conformational changes within the translocation domain (H_N_) that grants entry of the catalytic domain (LC) into the cytosol. Upon entry, the LC, a Zn^2+^ dependent endopeptidase, can cleave its target SNARE protein (SNAP-25 for BoNT/A) preventing vesicular fusion and neurotransmitter release [19].

Gangliosides consist of a hydrophilic oligosaccharide moiety containing sialic acid and a hydrophobic lipid tail that is embedded in the cell membrane of most vertebrate cells [20]. They are more abundant on the surface of nerve cells and have been identified in a range of biochemical processes such as cell–cell recognition and signal transduction [21]. GD1a (Figure 1A) constitutes one of the four major gangliosides (GM1, GD1a, GT1b and GD1b [22]) that make up 80–90% [20] of all gangliosides. The oligosaccharide moiety contains six monosaccharide units, of which the three terminal units (Sialic acid, Galactose, and *N*-acetylglucosamine) have been shown to form direct hydrogen bonding interactions with the H_CC_ subdomain of BoNT subtypes (/A1 to/A5) [23,24,25,26,27]. The terminal units are conserved among GD1a and GT1b and both have been identified as binding partners with BoNT/A1 [28].

Here, we report the high-resolution structure of BoNT/A6 cell binding domain in complex with GD1a (H_C_/A6:GD1a), and a new crystal form of H_C_/A6 [H_C_/A6 (crystal form II)]. A detailed analysis of these two structures along with a previous H_C_/A6 structure [H_C_/A6 (crystal form I)] [29], reveals the interactions that occur across the H_C_/A6:GD1a interface, and the conformational flexibility of H_C_/A6. The structural information presented here may aid the development of novel BoNT-based therapeutics and our understanding of BoNT function.

## 2. Results and Discussion

### 2.1. New Crystal Form of H_C_/A6

During the screening of H_C_/A6 co-crystallisation with GD1a we identified the presence of two crystal forms in one drop. One form was identical to a previously published H_C_/A6 structure, that we refer to as H_C_/A6 (crystal form I; PDB code 6TWO [29]), which belonged to the space group P2_1_2_1_2_1_ with unit cell dimensions of a = 39.54 Å, b = 105.59 Å, and c = 112.41 Å [29]. The overall fold is near-identical to other H_C_/A subtypes, consisting of an N-terminal 14 β-strand ‘jelly roll’ fold and a C-terminal ‘β-trefoil’. The GBS also has high structural similarity to H_C_/A5 [29]. However, in the new crystal form [referred here as H_C_/A6 (crystal form II)] the ‘*b*’ crystal lattice parameter differed by 33.47 Å with unit cell dimensions of a = 39.55 Å, b = 78.94 Å and c = 118.55 Å (Table 1). H_C_/A6 (crystal form II) required two molecular replacement (MR) search models (individual H_CN_ and H_CC_ subdomains) for correct placement of H_C_/A6 molecule in the electron density map. It was possible to model in the β-sandwich portion of H_CN_ and there was clear electron density throughout the H_CC_ subdomain, except for 12 residues (1140–1151) within the SV2 binding loop (1139–1157). Although there is variation in crystal contacts between H_C_/A6 (crystal form I) and H_C_/A6 (crystal form II), both structures have the ganglioside binding site (GBS) occluded by Lys 1170 from a symmetry-related molecule (Figure 1B), hence GD1a binding was not observed. H_C_/A6 (crystal form I) has a higher solvent content than H_C_/A6 (crystal form II) (46.87% and 33.58%, respectively) and the interface between the H_CN_ and H_CC_ subdomain is larger (858.7 Å^2^ and 461.1 Å^2^, respectively).

### 2.2. Crystal Structure of H_C_/A6 in Complex with GD1a Oligosaccharide

The crystal structure of H_C_/A6:GD1a complex was determined at 1.5 Å resolution by molecular replacement in the monoclinic space group P2_1_ with one molecule in the asymmetric unit (Table 1). There was clear electron density at the expected ganglioside binding site (GBS) for which 5 out of the 6 GD1a monosaccharides could be modelled (Figure 1C, blue).

The overall fold of H_C_/A6 did not change upon binding to GD1a, as indicated by a low RMSD (0.81 Å for C_α_ atoms) between H_C_/A6:GD1a and H_C_/A6 (crystal form I) (PDB code 6TWO [29]) structures. However, at the GBS, residue Phe 1278 rotates towards Sia^5^ and the loop 1269–1277 appears to widen by 4 Å (measured by the difference in C_α_ position between Arg 1269 and Thr 1277 residues) (Figure 1C, magenta). This was observed previously for H_C_/A2, H_C_/A3, and H_C_/A5. Furthermore, H_C_/A6 forms seven hydrogen bonds with GD1a at Sia^5^, Gal^4^, and GalNAc^3^ through six residues that are conserved among all BoNT/A subtypes (Figure 2) [23,24,25,26,27]. The H_C_/A6:GD1a interface is most similar to H_C_/A5, forming a total of 7 hydrogen bonds with the three monosaccharides GalNAc^3^, Gal^4^ and Sia^5^ [27].

### 2.3. HC/A6 (Crystal Form II) Reveals a Large Hinge-Rotation between HCN and HCC Subdomains

Superimposition of H_C_/A6 (crystal form I) (PDB code 6TWO [29]) and H_C_/A6:GD1a structures with H_C_/A6 (crystal form II) (RMSD values of 2.78 and 2.92 Å, respectively, for C_α_ atoms) revealed a misalignment in C_α_ positioning across the entire molecule (Figure 3A). The program Dyndom [30] revealed a large hinge rotation of ~16.8° in H_C_/A6 (crystal form II) when compared to H_C_/A6 (crystal form I) and H_C_/A6:GD1a (Figure 4). To date, this is the largest “hinge motion” in subdomain orientation observed among BoNT/A subtype structures [25,27], and it suggests a high degree of flexibility existing between the H_CN_ and H_CC_ subdomains. The biological implication of this hinge-rotation has not yet been determined; however, it has been previously suggested that it may aid in orientating the H_N_ towards the membrane in preparation for translocation [23].

### 2.4. Structural Comparison of HC/A6 in the Presence/Absence of GD1a

To determine the local structural differences between the H_C_/A6 (crystal forms I & II) and H_C_/A6:GD1a, the H_CC_ and H_CN_ subdomains were superimposed independently (Figure 3B,C)—this revealed two regions of conformational flexibility within the H_CC_ subdomain (Figure 3B). The first is within the SV2 binding site (1139–1157) where residues 1150–1152 in both H_C_/A6:GD1a and H_C_/A6 (crystal form I) structures form a β-sheet with residues 1003–1006. However, for the H_C_/A6 (crystal form II) structure, this β-sheet could not be modelled due to disorder, possibly due to the large hinge motion between the two subdomains separating those residues. This appears to be consistent with residues 1152–1157 being rotated away from the H_CN_ subdomain when compared to H_C_/A6:GD1a and H_C_/A6 (crystal form I), accompanied by a flip at Tyr 1155 (Figure 3B, magenta). This likely contributes to the large difference in interface surface area observed between the H_CN_ and H_CC_ subdomains. The flexibility of the SV2 binding site might act to promote anchoring of H_C_/A6 to SV2, as the binding loop is able to extend outward away from the protein and sample more conformational space.

The second structural difference is within a polar/charged loop (residues 1225–1236) positioned close to the GBS (Figure 3B). This loop adopts alternative conformations in H_C_/A6 (crystal form I) and H_C_/A6 (crystal form II) but is only partially modelled in H_C_/A6:GD1a (Figure 3B, blue). A comparison of residues Cys 1235, Lys 1236, and Cys 1280 reveals an alternating bridging interaction (Figure 3B, blue). For the H_C_/A6 (crystal form II) structure there is a continuation of electron density between the Nζ of Lys 1236 and Sγ of Cys 1280 residues (Figure 3D), whereas in the H_C_/A6:GD1a structure Cys 1280 forms a disulphide bridge with Cys 1236, and the structure of H_C_/A6 (crystal form I) shows neither (Figure 3, blue). This is consistent with similar observations with H_C_/A2 and H_C_/A5 [25,29] and such Lys-O-Cys bridging interactions are believed to be widespread in protein structures but under-reported [31]. It is likely that these different bridging interactions could lead to multiple conformations that the loop can adopt, however, the biological significance of the Cys-Lys bridge remains unclear at the moment.

### 2.5. B-Factor Analysis

To further assess the conformational flexibility of H_C_/A6 we used the program BANΔIT (a tool for the normalisation and analysis of B-factor profiles [32]) to produce a raw (Figure 5A) and normalised (Figure 5B) B-factor plot for the H_C_/A6:GD1a and H_C_/A6 (crystal form II) structures. B-factors quantify the relative motion of individual atoms within a crystal structure, where an increase in B-factor indicates an increase in motion, providing insights into the flexible regions of a protein. The raw plot shows higher B-factors overall for the H_C_/A6 (crystal form II) structure, indicating more flexibility across the entire molecule compared to the H_C_/A6:GD1a structure. For the latter, the H_CC_ subdomain is more flexible than the H_CN_ (Figure 5B). This is consistent with the observed conformational flexibility of the H_CC_ subdomain, perhaps to accommodate binding to both SV2 and a ganglioside on the surface of motor neurons.

## 3. Materials and Methods

### 3.1. Protein Expression and Purification

The cell binding domain (residues 871–1296) of BoNT/A6 (H_C_/A6) was previously cloned into the pJ401 vector [29] and subsequently transformed into BL21 (DE3) star cells. Expression cultures were grown at 37 °C until an OD_600_ of 0.6. The culture was induced with 1 mM IPTG and incubated overnight at 16 °C. Cells were harvested by centrifugation and the pellet was resuspended in 50 mM Tris pH 7.4, 0.5 M NaCl. Cells were then lysed by homogenisation in a cell disrupter and the lysate was clarified by centrifugation. H_C_/A6 was captured using Ni^2+^ affinity chromatography and eluted with a 0–0.5 M imidazole linear gradient (in 50 mM Tris pH 7.4, 0.5 M NaCl). H_C_/A6 was further purified by gel filtration in 50 mM Tris pH 7.2, 150 mM NaCl using a GE superdex 200 column. H_C_/A6 was concentrated to 1 mg/mL and flash frozen in liquid nitrogen for storage at −20 °C until required for crystallisation.

### 3.2. X-ray Crystallography

H_C_/A6 was concentrated to 6 mg/mL and incubated with 5 mM GD1a for 1 h at room temperature prior to setting up crystallisation screens using the sitting drop vapour diffusion method in Swissci Intelli Well plates (High Wycombe, UK) at 16 °C. Crystals were identified in the PACT premier and BCS screens supplied by Molecular Dimensions (Rotherham, UK). The best H_C_/A6:GD1a crystals grew in 0.2 M NaCl, 0.1 M HEPES pH 7.0, 20% *w*/*v* PEG 6000. H_C_/A6 crystals (crystal form II) grew in 0.1 M magnesium acetate tetrahydrate, 0.1 M MES pH 6.5, 12% *w*/*v* of 50% PEG smear medium (4.55% *v*/*v* PEG 400, 4.55% *v*/*v* PEG 500 MME, 4.55% *v*/*v* PEG 600, 4.55% *w*/*v* PEG 1000, 4.55% *w*/*v* PEG 2000, 4.55% *w*/*v* PEG 3350, 4.55% *w*/*v* PEG 4000, 4.55% *w*/*v* PEG 5000 MME, 4.55% *w*/*v* PEG 6000, 4.55% *w*/*v* PEG 8000, 4.55% *w*/*v* PEG 10,000), 10% *v*/*v* ethylene glycol. Crystals were mounted using a cryo-loop and flash frozen in liquid nitrogen. A total of 7200 images were collected on I04 at Diamond Light Source (Didcot, UK) at 0.1° for 0.01 s per image. Indexing and integration of X-ray diffraction data were performed using DIALS [33]. Subsequent data processing was performed using the CCP4 software suite [34]. Data reduction and merging was performed in AIMLESS as part of CCP4 [34]. The initial phases were determined by molecular replacement using Phaser [35]. H_C_/A6:GD1a was determined using H_C_/A6 (crystal form I) as a single search model. H_C_/A6 (crystal form II) was determined using the H_CN_ and H_CC_ subdomains as separate search models from the H_C_/A6 (crystal form I) structure (PDB code 6TWO [29]). Both structures were refined using REFMAC [36] and Phenix [37] and molecular modelling was performed in COOT [38], The structures were validated using Molprobity [39] and PDB validation. All figures were produced using CCP4mg [40].

## 4. Conclusions

The crystal structure of H_C_/A6:GD1a revealed six residues of H_C_/A6 which form seven hydrogen bonding interactions with the three terminal monosaccharides of GD1a. Overall, the conformation of H_C_/A6 upon binding GD1a does not change dramatically except for a flipping of the Phe 1278 sidechain towards Sia^5^ of GD1a and an accompanying widening of loop 1269–1277. However, a new crystal form of H_C_/A6 (H_C_/A6 (crystal form II)) revealed a large 16.8° rotation between the H_CN_ and H_CC_ subdomain (hinge) that might aid membrane anchoring through both the SV2 and ganglioside binding site. Finally, a detailed comparison of the structures presented here (H_C_/A6:GD1A and H_C_/A6 (crystal form II)) with a previously reported H_C_/A6 structure (crystal form I) revealed the extent of conformational flexibility within the H_CC_ subdomain. Two areas of particular interest are the SV2 binding loop and a polar/charged loop close to the GBS. The SV2 loop, in the H_C_/A6 (crystal form II) structure, appears to have rotated away from the H_CN_ domain protruding outward from the surface of the protein. There is a polar/charged loop that adopts a different conformation in each structure along with a dynamic bridging interaction that alternates between Cys 1235-Cys 1280, Lys 1236-O-Cys 1280, and no bridge at all. The biological implication of these structural features is yet to be established and will require further experimental investigation.

## Figures and Tables

**Figure 1 ijms-23-09620-f001:**
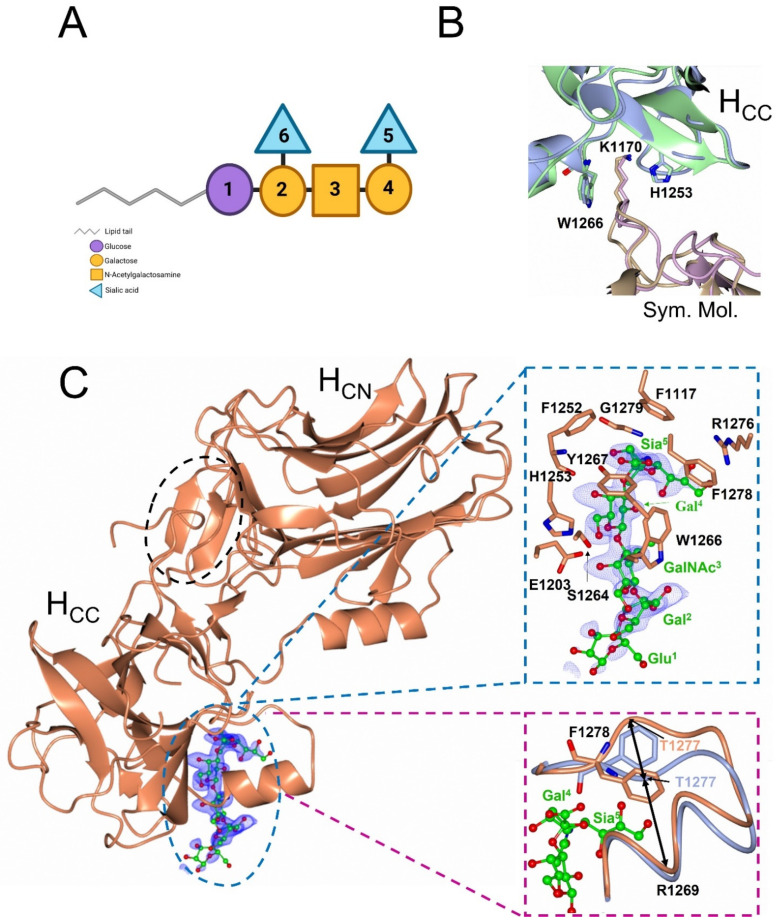
(**A**) GD1a ganglioside schematic, created with Biorender.com. 1 = Glucose, 2/4 = Galactose, 3 = N-Acetylgalactosamine, 5/6=Sialic acid. (**B**) The ganglioside binding site of H_C_/A6 (crystal form I) (grey) and H_C_/A6 (crystal form II) (green) is occluded by Lys 1170 of a symmetry molecule (pink/cream). (**C**) Crystal structure of *Clostridium botulinum* neurotoxin A6 cell binding domain in complex with GD1a (orange). The dashed blue inset shows how GD1a fits into the electron density (F_O_-Fc, contoured at 3σ level) in the ganglioside binding site (GBS) and surrounding residues. The SV2 binding site is indicated by the black dashed oval. The dashed magenta inset highlights the conformational changes associated with GD1a binding by comparison of H_C_/A6 (crystal form I) (grey) with H_C_/A6:GD1a (orange), where the loop of residues 1269–1277 widens upon binding and Phe 1278 flips towards the Sia^5^ moiety of GD1a ganglioside. The large arrows indicate the widening of the loop as measured by the change in Cα distance between R1269 and T1277.

**Figure 2 ijms-23-09620-f002:**
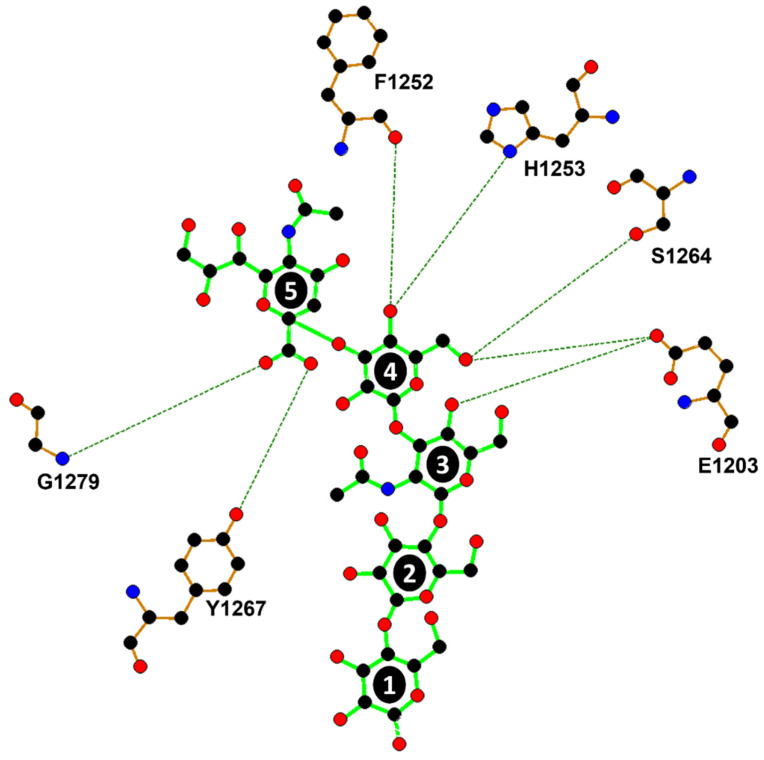
Ligplot^+^ of H_C_/A6 and GD1a hydrogen bonding interactions. 1 = Glc^1^, 2 = Gal^2^, 3 = GalNAc^3^, 4 = Gal^4^, 5 = Sia^5^. Sia^6^ is unmodelled due to insufficient electron density. Red=oxygen atom, blue=nitrogen atom, black=carbon atom. GD1a bonds are displayed in green, H_C_/A6 bonds in brown, and hydrogen bonds by dotted lines.

**Figure 3 ijms-23-09620-f003:**
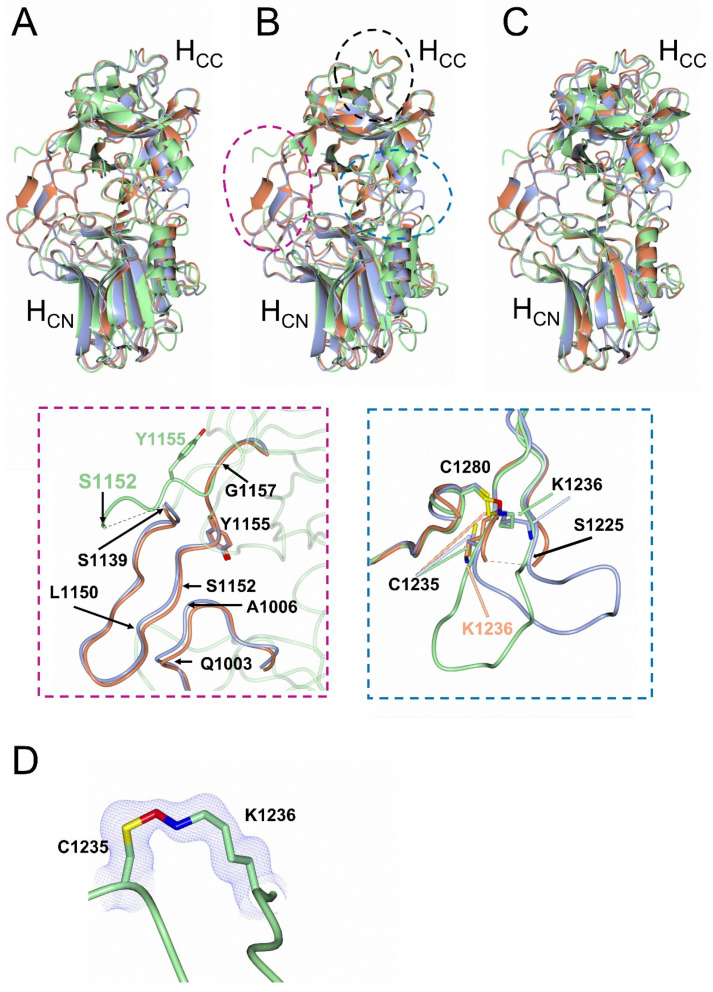
Structural comparison of H_C_/A6 (crystal form I), H_C_/A6 (crystal form II), and H_C_/A6:GD1a. (**A**) Overall superimposition of H_C_/A6 (crystal form I) (grey) (PDB code 6TWO [29]), H_C_/A6 (crystal form II) (green), and H_C_/A6:GD1a (orange). Superimposition of H_CC_ (**B**) and H_CN_ (**C**) subdomains separately. The GBS is highlighted by a dashed black oval, the dashed magenta oval highlights the structural differences at the SV2 binding site (dashed magenta inset), and the dashed blue oval highlights the structural differences at the polar/charged loop of residues 1225–1236. In H_C_/A6 (crystal form I) we observe no bridging interaction involving Cys 1280, whereas in H_C_/A6:GD1a and H_C_/A6 (crystal form II) Cys 1280 forms a disulphide bridge with Cys 1235 and a Lys-O-Cys bridge with Lys 1236, respectively (dashed blue inset). (**D**) The 2F_O_-F_C_ electron density map contoured at 1 σ level for the Lys-O-Cys bridge present in the H_C_/A6 (crystal form II) structure.

**Figure 4 ijms-23-09620-f004:**
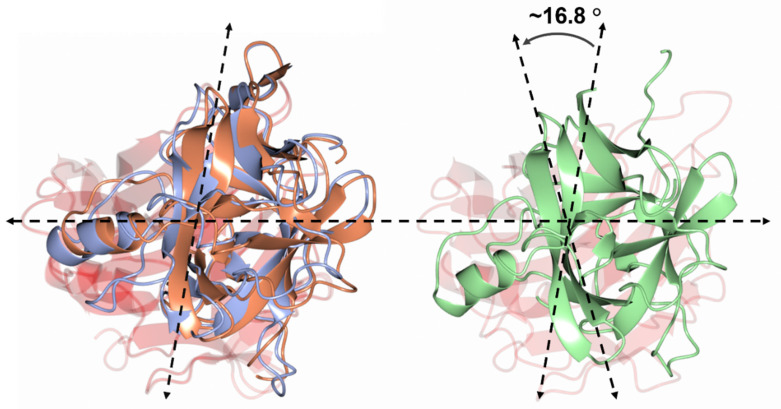
Subdomain hinge rotation. The position of the H_CN_ subdomain position of H_C_/A6 (crystal form II) (green) is different to that H_C_/A6 (crystal form I) (grey) (PDB code 6TWO [29]) and H_C_/A6:GD1a (orange) when their H_CC_ subdomains (transparent red) are superimposed. There appears to be a rotation of 16.8° between the H_CN_ subdomains as calculated by Dyndom [30].

**Figure 5 ijms-23-09620-f005:**
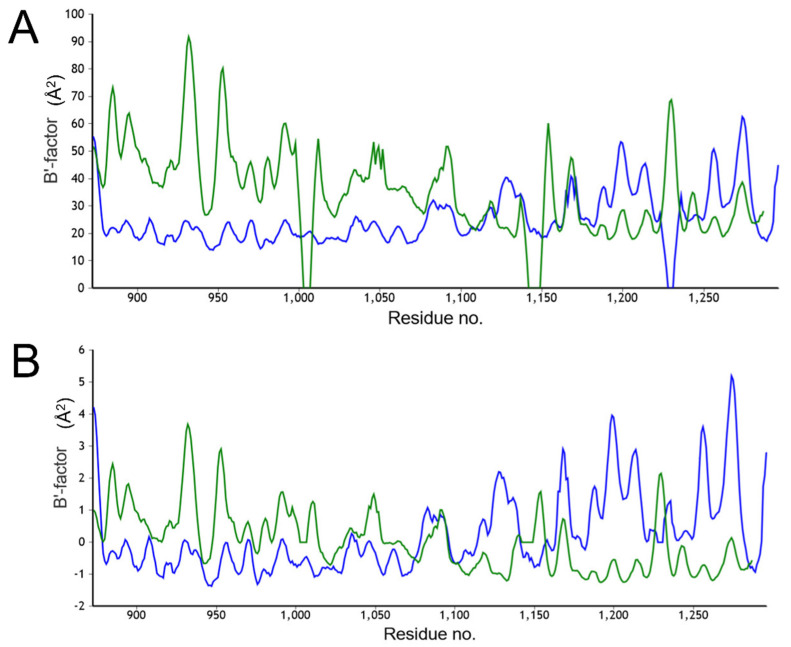
B-factor analysis of H_C_/A6:GD1a and H_C_/A6 (crystal form II). BANΔIT was used to produce a raw (**A**) and normalised (**B**) B-factor plot of H_C_/A6 (crystal form II) (green) and H_C_/A6:GD1a (blue) revealing the increased flexibility of the H_CC_ subdomain in the H_C_/A6:GD1a structure, as evidenced by an increase in relative B-factors.

**Table 1 ijms-23-09620-t001:** X-ray crystallographic data collection and refinement statistics for the structures of H_C_/A6 in complex with GD1a (H_C_/A6:GD1a) and the new crystal form of H_C_/A6 [H_C_/A6 (crystal form II)]. Outer shell statistics are in parenthesis.

Beamline	I04	I04
Wavelength	0.9795 Å	0.9795 Å
Protein	H_C_/A6:GD1a	H_C_/A6 (crystal form II)
**Crystallographic statistics**		
Space group	P2_1_	P2_1_2_1_2_1_
Unit cell dimensions a, b, c (Å) α, β, γ (°)	44.31, 83.64, 58.13 90.00, 98.66, 90.00	39.55, 78.94, 118.55 90.00, 90.00, 90.00
Resolution range (Å)	57.65–1.50 (1.53–1.50)	118.55–1.50 (1.53–1.50)
R_merge_	0.225 (8.03)	0.119 (5.61)
R_pim_	0.063 (2.24)	0.023 (1.13)
<I/σ(I)>	7.2 (0.3)	12.0 (0.7)
CC1/2	0.998 (0.991)	1.00 (0.33)
Completeness (%)	100.0 (100.0)	99.6 (97.5)
No. observed reflections	914,657 (45,756)	1,580,456 (75,049)
No. unique reflections	67,218 (3357)	60,147 (2894)
Multiplicity	13.6 (13.6)	26.3 (25.9)
**Refinement Statistics**		
Rwork/Rfree	0.204/0.235	0.248/0.289
RMSD bond lengths (Å)	0.010	0.0098
RMSD bond angles (°)	1.542	1.546
Ramachandran plot statistics (%) Favoured Allowed Outliers	96.00 4.00 0.00	93.00 7.00 0.00
Average B-Factors (Å2) Protein atoms Solvent atoms GD1a ligand	28.10 32.00 52.30	38.87 38.05 N/A
No. Atoms Protein Solvent GD1a	3734 3423 244 68	3286 3127 159 N/A
PDB code	8AGK	8ALP

## Data Availability

The atomic coordinates and structure factors of H_C_/A6:GD1a and H_C_/A6 (Crystal form II) have been deposited in the protein data bank under the accession codes 8AGK and 8ALP, respectively.

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
