# Peer review of "Crystal Structures of the Clostridium botulinum Neurotoxin A6 Cell Binding Domain Alone and in Complex with GD1a Reveal Significant Conformational Flexibility"

_ijms, 2022, doi:10.3390/ijms23179620_

Round 1

Reviewer 1 Report

The manuscript “Crystal Structures of the Clostridium botulinum Neurotoxin A6 Cell Binding Domain Alone and in Complex with GD1a Reveal Significant Conformational Flexibility” is focused on the structural investigation of the botulinum neurotoxin A6 cell binding domain (HC/A6) and of its complex with GD1a. The authors describe the interactions involved in ganglioside binding and the characterization of a new crystal form of HC/A6, named crystal form II, in which they observed a large ‘hinge motion’ between the HCN and HCC subdomains, revealing the degree of conformational flexibility exhibited by HC/A6. The manuscript provides significant insights on the target and I would recommend its publication after addressing the minor issues reported below.

Minor issues:

1. Please correct the format of multiple references throughout the manuscript (e.g., at lines 39, 41, 45).

2.     Section 1, line 47. Change “or endogenous” in “or an endogenous”.

3.     Section 1, line 54. Change “and ganglioside” in “and a ganglioside”.

4.  Section 1, line 64. A figure of the GD1a chemical structure could be added here to help the reader in visualizing its structure.

5.  Section 1, lines 75-77. This sentence provides details on the results achieved by the authors and it could be removed from the introduction of the manuscript.

6.     Section 1, line 78. “in general” should be removed.

7. Section 2.1, line 84. Change “had a space group of P212121” in “belonged to the space group P212121

8.  Section 2.1, line 86. Change “b cell dimension” in “b crystal lattice parameter”.

9.  Section 2.1, lines 87-97. The main structural features of the HC/A6 crystal form I should be briefly described before reporting the differences observed with the crystal form II. Furthermore, the authors observe a variation in crystal contacts between HC/A6 (crystal form I) and HC/A6 (crystal form II), reporting that both structures have the ganglioside binding site (GBS) occluded by Lys1170 from a symmetry-related molecule, preventing GD1a binding. This information should be supported by a figure showing these structural features.

10.  Table 1. The presence of outliers in the Ramachandran plot should be briefly explained, maybe during the structural description in Section 2.1.

11.  Figure 1. The authors should display the omit map in the figure and report the contouring level in the figure caption.

12. Section 2.2, lines 117-118. It is not clear how the authors have measured the widening of the loop, please clarify.

13.  Section 2.3, lines 126-129. The sentence is not clear, please clarify.

14.  Figure 3 is not clear, the authors should present crystal structures in cartoon and enlarge the A-C panels showing structural comparisons. 

15.  Section 4.2, line 232. “of” should be removed.

16.  Section 4.2, line 242. Please correct “Oxon”.

Author Response

  1. Please correct the format of multiple references throughout the manuscript (e.g., at lines 39, 41, 45).

Have corrected with track changes

  1. Section 1, line 47. Change “or endogenous” in “or an endogenous”.

Have corrected with track changes

  1. Section 1, line 54. Change “and ganglioside” in “and a ganglioside”.

Have corrected with track changes

  1. Section 1, line 64. A figure of the GD1a chemical structure could be added here to help the reader in visualizing its structure.

Added to figure 1

  1. Section 1, lines 75-77. This sentence provides details on the results achieved by the authors and it could be removed from the introduction of the manuscript.

Corrected

  1. Section 1, line 78. “in general” should be removed.

Have corrected with track changes

  1. Section 2.1, line 84. Change “had a space group of P212121” in “belonged to the space group P212121

Have corrected with track changes

  1. Section 2.1, line 86. Change “b cell dimension” in “b crystal lattice parameter”.

Corrected with track changes

Corrected

  1. Section 2.1, lines 87-97. The main structural features of the HC/A6 crystal form I should be briefly described before reporting the differences observed with the crystal form II. Furthermore, the authors observe a variation in crystal contacts between HC/A6 (crystal form I) and HC/A6 (crystal form II), reporting that both structures have the ganglioside binding site (GBS) occluded by Lys1170 from a symmetry-related molecule, preventing GD1a binding. This information should be supported by a figure showing these structural features.

Have added a small description of the overall fold and the figure.

  1. Table 1. The presence of outliers in the Ramachandran plot should be briefly explained, maybe during the structural description in Section 2.1.

This was ‘1’ from a previous draft and has been corrected to zero as per the final validation report.

  1. Figure 1. The authors should display the omit map in the figure and report the contouring level in the figure caption.

2Fo-Fc map changed to Fo-Fc map instead in figure 1

  1. Section 2.2, lines 117-118. It is not clear how the authors have measured the widening of the loop, please clarify.

Annotated figure further to indicate loop widening

  1. Section 2.3, lines 126-129. The sentence is not clear, please clarify.

Have addressed with track changes

  1. Figure 3 is not clear, the authors should present crystal structures in cartoon and enlarge the A-C panels showing structural comparisons. 

Corrected with track changes, B and C swapped due to amended figure

  1. Section 4.2, line 232. “of” should be removed.

Corrected with track changes

  1. Section 4.2, line 242. Please correct “Oxon”.

Have corrected with track changes

Reviewer 2 Report

The manuscript by Gregory et al. describes the crystal structures of Clostridium Botulinum Neurotoxin A6 cell binding domain in the apo form and in complex with ganglioside GD1a. Both structures were solved at high resolution (1.5 A). Overall, this is a well written paper and the structure determination part is well described. However, there is a big short coming with the paper as it fails to compared their results with previous structures. There is a previously reported structure of the BoNTHc in the same two crystal forms and the authors should comment on this. More importantly there are several structures of othe BoNT-Hc in complex with GD1a and Sialyl-T, it is critical that they analyse their results and comment on general feautures of how the different BoNT recognize gangliosides. It is surprising that giving the wealth of structural information available, the authors barely comment on it.

Author Response

Have added a sentence at the end of section 2. 2 to address this point without loosing the main focus of our current manuscript.  The general features of how different BoNT molecules recognize gangliosides (based on the wealth of structural data will form a Review topic for the future) as more structural details are becoming available.

Reviewer 3 Report

No comments or suggestions

Author Response

No comments to address.